# Quality and sustainability of Ethiopia's national surgical indicators

**Kayleigh R. Cook**[1,☉], **Zebenay B. Zeleke**[2☉], **Ephrem Gebrehana**[3], **Daniel Burssa**[4], **Bantalem Yeshanew**[4], **Atkilt Michael**[5], **Yoseph Tediso**[5], **Taylor Jaraczewski**[1], **Chris Dodgion**[1], **Andualem Beyene**[6‡], **Katherine R. Iverson**[1‡]*

**1** Department of Surgery, Medical College of Wisconsin, Milwaukee, Wisconsin, United States of America, **2** Department of Surgery, Bahir Dar University, Bahir Dar, Amhara Region, Ethiopia, **3** Department of Surgery, Hawassa University, Hawassa, Sidama Region, Ethiopia, **4** Ethiopian Ministry of Health, Addis Ababa, Ethiopia, **5** Sidama Regional Health Bureau, Hawassa, Sidama Region, Ethiopia, **6** Department of Surgery, Addis Ababa University, Addis Ababa, Ethiopia

☉ These authors contributed equally to this work.
‡ AB and KRI also contributed equally to this work.
* katie.r.iverson@gmail.com

**Data Availability Statement:** For this study, anonymized data for review has been uploaded as a supporting information file titled "S1 Data." This data was shared with the study investigators with

## Abstract

In 2015, the Ethiopian Federal Ministry of Health (FMOH) developed the Saving Lives through Safe Surgery (SaLTS) initiative to improve national surgical care. Previous work led to development and implementation of 15 surgical key performance indicators (KPIs) to standardize surgical data practices. The objective of this project is to investigate current practices of KPI data collection and assess quality to improve data management and strengthen surgical systems. The first portion of the study documented the surgical data collection process including methods, instruments, and effectiveness at 10 hospitals across 2 regions in Ethiopia. Secondly, data for KPIs of focus [1. Surgical Volume, 2. Perioperative Mortality Rate (POMR), 3. Adverse Anesthetic Outcome (AAO), 4. Surgical Site Infection (SSI), and 5. Safe Surgery Checklist (SSC) Utilization] were compared between registries, KPI reporting forms, and the DHIS2 (district health information system) electronic database for a 6-month period (January—June 2022). Quality was assessed based on data completeness and consistency. The data collection process involved hospital staff recording data elements in registries, quality officers calculating KPIs, completing monthly KPI reporting forms, and submitting data into DHIS2 for the national and regional health bureaus. Data quality verifications revealed discrepancies in consistency at all hospitals, ranging from 1–3 indicators. For all hospitals, average monthly surgical volume was 57 cases, POMR was 0.38% (13/3399), inpatient SSI rate was 0.79% (27/3399), AAO rate was 0.15% (5/3399), and mean SSC utilization monthly was 93% (100% median). Half of the hospitals had incomplete data within the registries, ranging from 2–5 indicators. AAO, SSC, and SSI were commonly missing data in registries. Non-standardized KPI reporting forms contributed significantly to the findings. Facilitators to quality data collection included continued use of registries from previous interventions and use of a separate logbook to document specific KPIs. Delayed rollout of these indicators in each region contributed to issues in data quality. Barriers involved variable indicator recording from different personnel, data collection tools that generate false positives (i.e. completeness of SSC defined as paper form filled out prior to

the permission of the Ethiopian Federal Ministry of Health, the relevant Regional Health Bureaus for the regions described, and the participating hospitals. The names of the specific hospitals have been removed for confidentiality purposes in compliance with ethics review boards. The Ethiopian Federal Ministry of Health retains the rights to this data and therefore prevents additional release of data, beyond what is provided in the supporting information file. For more information, please contact Dr. Abiy Dawit, SALTS II, Ethiopian Federal Ministry of Health at abiy.dawit@moh.gov.et.

**Funding:** Study author Cook KR received travel funding through the Dr. Elaine Kohler Summer Academy of Global Health Research Scholarship, funded by the Wm. Collins Kohler Foundation. URL link: https://www.mcw.edu/departments/office-of-global-health/research. This funder had no role in study design, data collection and analysis, decision to publish, or preparation of the manuscript. The other authors received no specific funding for this work.

**Competing interests:** The authors have declared that no competing interests exist.

patient discharge) or missing data because of reporting time period (i.e. monthly SSI may miss infections outside of one month), inadequate data elements in registries, and lack of standardized monthly KPI reporting forms. As the FMOH introduces new indicators and changes, we recommend continuous and consistent quality checks and data capacity building, including the use of routinely generated health information for quality improvement projects at the department level.

## Introduction

Creating effective global surgery programs requires a clear understanding of the baseline state of surgical care. While the global surgery movement has gained momentum in the past several years, there remains a lack of quality data regarding the status of surgery in low-resource settings [1]. Our understanding of surgical capacity, resources, and outcomes has been limited by our methods of inquiry and the system in place for measuring results.

The quest for global surgery data was instigated by the Lancet Commission on Global Surgery [2], who proposed six core indicators designed to measure access to safe and affordable surgical care. A number of these were integrated into the World Bank's World Development Indicators [3] and the World Health Organization's Global Health Indicators [4]. Based on expert opinion in the Utstein consensus report, the global surgery indicators have since been uniformly defined and narrowed to five (surgical volume, perioperative mortality rate (POMR), surgical workforce, financial risk protection, and geospatial access) [5]. Information on these metrics have come from a variety of methodological approaches in the global community. Literature reviews, modeling studies, facility-based surveys, and regional and international surgical outcomes collaboratives are the diverse sources for these data [6–14]. While these studies have provided the foundation for global surgery indicator benchmarking, there is a growing need for sustainable and timely country-based systems with surgical metrics integrated into national health information systems [15, 16].

A monitoring and evaluation pillar was included in Ethiopia's national surgical strategy (SaLTS or Saving Lives Through Safe Surgery) to gauge the current surgical practice in the country and create a sustainable method for tracking progress [17, 18]. The surgical key performance indicators (KPIs) are 15 metrics prospectively collected and reported regularly from each health facility in the country. In 2018, the indicators were piloted at 10 hospitals across two regions of Ethiopia as part of Safe Surgery 2020, an initiative to improve surgical care in the country [19]. Since 2018, the electronic platform DHIS2 has been used to report and aggregate these indicators. Informed by the evaluation of the first five-year surgical strategic plan (SaLTS) in 2020, the Ethiopian Federal Ministry of Health has now initiated the second stage of this strategy, SaLTS II, in 2021. The SaLTS II plan [20] outlines the approach to meet surgical metric goals by the year 2025, that include increased surgical volume and providers in line with international standards, POMR < 2%, and improved access to care within two-hours. To evaluate the sustainability of the SaLTS data system and ensure its utility in monitoring progress towards these goals, this study was created to appraise these key surgical indicators.

The aim of this study is to investigate the data management practices and assess the quality of surgical data in Ethiopia by exploring how surgical indicators are collected and reported at the hospital level since their national implementation.

## Material and methods

### Study design

The study was designed as a multi-institution, retrospective record review with hospital visits as the main source of program evaluation. A total of ten hospitals, split evenly between the Amhara and Sidama regions, were visited by the research team from July-August 2022. During hospital visits, surgical data practices were observed to assess the flow of information between hospital registries, hospital monthly key performance indicator (KPI) forms, and the District Health Information Software 2 (DHIS2) reports reviewed at the regional and national level. Data owners and instruments were identified at each step. The dates of record review ranged from 1/1/2022 through 6/31/2022. This time period was chosen to evaluate the most recent six-months of hospital data from the time of inquiry. All data accessed was deidentified and the study authors were unable to identify specific individuals from the data. The research team undertook a complete surgical data audit at each hospital visited. Registries within the OR, surgical ward, ICU, and maternity ward were reviewed by each reporting period and data points were aggregated to calculate monthly indicators. Researchers met with hospital quality improvement officers to audit monthly KPI reporting forms and final DHIS2 electronic database reports. At the Regional Health Bureaus, the final data reports from the queried hospitals visited were reviewed. At the national level, the DHIS2 conglomerate data from each Regional Health Bureau was evaluated. Data triangulation compared hospital to regional to national data.

### Subjects

Primary and general public hospitals who were previously involved in Safe Surgery 2020 interventions, with approval from the Amhara and Sidama Regional Health Bureau and the Ministry of Health, were selected for this research [19, 21]. Amhara hospitals were further chosen based on their participation in the 2018 surgical data intervention carried out by the research team. Sidama hospitals were selected according to their involvement in Safe Surgery 2020 programming. Records of patients who underwent major surgery, defined as any procedure conducted in an operating room under general, spinal, or regional anesthesia [22], during their hospital stay within the study period were reviewed in this study.

### Methods of measurements

Surgical Key Performance Indicators (KPIs) of interest were selected based on a prior surgical data intervention conducted at Amhara hospitals and on feedback from the Federal Ministry of Health (FMOH). The research team selected five indicators: Surgical Volume, Perioperative Mortality Rate (POMR), Rate of Safe Surgery Checklist Utilization (SSC), Surgical Site Infection Rate (SSI), and Anesthetic Adverse Outcome Rate (AAO) for data analysis. Formal definitions for each indicator were defined by the Saving Lives Through Safe Surgery (SALTS) program [22] and are included in Table 1. For collection of data from the registries, KPI reporting forms, and DHIS2 data, the investigators utilized Microsoft Excel forms at each hospital during on-site visits.

### Data analysis

Enumerated data from hospital registries were compared with the data in the national database. Analysis consisted of comparing data between sources (registry, KPI reporting forms, and DHIS2) and assessing data quality according to dimension 1 (data completeness) and dimension 2 (internal consistency of reported data) of the Data Quality Review (DQR)

**Table 1. Five surgical key performance indicators of focus and definitions [19].**

| Indicator | Definition |
|---|---|
| Surgical Volume | The total number of major surgical procedures performed in an operating theater per 100,000 population per year. Note: A major surgical procedure is defined as any procedure conducted in an OR under general, spinal, or major regional anesthesia. |
| Perioperative Mortality Rate (POMR) | The all-cause death rate before discharge among patients who underwent a major surgical procedure in an operating theatre during the reporting period. Note: Stratified by emergent and elective major procedures |
| Rate of Safe Surgery Checklist Utilization | The proportion of surgical procedures where the safe surgery checklist was fully implemented. |
| Surgical Site Infection (SSI) Rate | The proportion of all major surgeries with an infection occurring at the site of the surgical wound prior to discharge. One or more of the following criteria should be met: Purulent drainage from the incision wound Positive culture from a wound swab or aseptically aspirated fluid or tissue Spontaneous wound dehiscence or deliberate wound revision/opening by the surgeon in the presence of: pyrexia > 38C or localized pain or tenderness Wound pain, tenderness, localized swelling, redness or heat AND incision opened by the surgeon or spontaneously dehisced Note: A major surgical procedure is defined as any procedure conducted in an OR under general, spinal, or major regional anesthesia. |
| Anesthetic Adverse Outcome Rate | The percentage of surgical patients who developed any one of the following: (1) Cardiorespiratory arrest, (2) High spinal anesthesia, or (3) Inability to secure airway. Cardiorespiratory arrest defined as: cessation of cardiac activity as evidenced by: Chest compressions being performed Loss of femoral, carotid, and apical pulse with ECG changes High spinal defined as: within 15 minutes of administration of spinal anesthesia: Patient experiences loss of sensation in the shoulder AND Need for positive pressure ventilation after administration of spinal anesthesia Includes any administration of spinal anesthesia extending above T4 level. Inability to secure airway defined as: Having to awaken patient due to inability to intubate Cardiac-respiratory arrest due to failure to intubate |

framework [22]. Completeness of data was evaluated by reviewing the monthly KPI reporting forms for the study period (Jan-June 2022) and documenting the number of months KPI data was absent for each indicator. For example, if the value for surgical volume was missing for one month at a single hospital, this was denoted as missing 1/6 of surgical volume data for the study period. Inconsistent data was defined as the number of months within the six-month study period where the registry source data did not match the DHIS2 electronic database. Following FMOH guidelines, surgical indicator data was deemed inconsistent if DHIS2 data differed by more than 10% of the registry data. This established verification factor for routine data quality assessment is calculated by the number of recorded events from the source document (registry) divided by the number of reported events in the DHIS2 report, with verification factor greater than 1.1 (under-reporting) or less than 0.9 (over-reporting) deemed inconsistent [23]. The KPI Safe Surgery Checklist utilization was an exception where KPI reporting forms were used as the source data and matched to DHIS2 data, given this indicator is not recorded in the registry.

## Ethical approval

Institutional review board approval was obtained from Medical College of Wisconsin (MCW) and the Ethiopian Public Health Institute (EPHI) for all study activities. EPHI approved the following protocol number: EPHI-IRB-454-2022, titled "Assessment of Current Data Practices

in Ethiopia." The MCW Institutional Review Board approved the study under PRO ID: PRO00043474. The study investigators also received a letter of support from the Federal Ministry of Health.

### Inclusivity in global research

Additional information regarding the ethical, cultural, and scientific considerations specific to inclusivity in global research is included in the S1 Checklist.

## Results

### Hospital characteristics

Table 2 includes hospital specific data on facility resources and characteristics for the ten study hospitals. Median catchment population was 375,000 patients (70,000–1,800,000) whereas median total hospital beds was 45 (25–145). Integrated Emergency Surgical Officers (IESOs), Master's-level health professionals intended to provide emergency and essential surgery in Ethiopia, were the predominant surgical provider at most institutions [24].

### Data collection processes

At all ten hospitals, data flow mechanisms were observed and compiled into a generalized process illustrated by Fig 1. The data collection process involved recording data elements in registries by hospital staff, then calculation of KPIs by quality improvement (QI) officers, completing monthly KPI reporting forms, and submitting data into the electronic national data system (DHIS2). For the five KPIs evaluated, healthcare providers entered surgical data into registries specific to their department, i.e. wards, ICU, or operating room. At the end of the reporting period, the QI officers reviewed the registries and utilized the raw data to calculate key performance indicators. If they noted any discrepancies or missing data, they would query the health provider responsible for that element. KPI data collection forms for this

**Table 2. Hospital-specific characteristics.**

| Sidama Region | | | | | | Amhara Region | | | | |
|---|---|---|---|---|---|---|---|---|---|---|
| | 1 | 2[a] | 3[a] | 4 | 5 | 6 | 7 | 8 | 9 | 10 |
| Catchment Population | 100K | 800K | 1.8M | 178K | 400K | 170K | 700K | 350K | 250K | 480K |
| Operating Rooms | 2 | 2 | 3 | 1 | 2 | 1 | 2 | 1 | 2 | 1 |
| Personnel: | | | | | | | | | | |
| IESOs | 0 | 2 | 5 | 2 | 1 | 4 | 3 | 4 | 4 | 4 |
| Surgeons | 1 | 1 | 5 | 1 | 1 | 1 | 0 | 1 | 1 | 0 |
| Anesthetists/ | 2 | 5 | 6 | 3 | 3 | 4 | 4 | 4 | 5 | 2 |
| Anesthesiologists | | | | | | | | | | |
| OB/GYNs | 0 | 2 | 3 | 0 | 0 | 1 | 0 | 0 | 1 | 0 |
| GPs | 11 | 22 | 57 | 12 | 13 | 20 | 16 | 21 | 15 | 12 |
| Beds: | | | | | | | | | | |
| Surgical | 4 | 12 | 27 | 4 | 6 | 13 | 11 | 16 | 13 | 7 |
| OB/GYN | 6 | 17 | 23 | 4 | 12 | 7 | 5 | 8 | 6 | 7 |
| Total in Hospital | 34 | 110 | 145 | 25 | 36 | 50 | 47 | 92 | 43 | 38 |

Sidama Region (Hospitals 1–5), Amhara Region (Hospitals 6–10). IESOs = Integrated Emergency Surgical Officers. OB = Obstetricians. GPs = General Practitioners. GYN = Gynecology.

[a]General hospitals considered secondary level health care. All other hospitals are primary hospitals, part of the primary level health care.

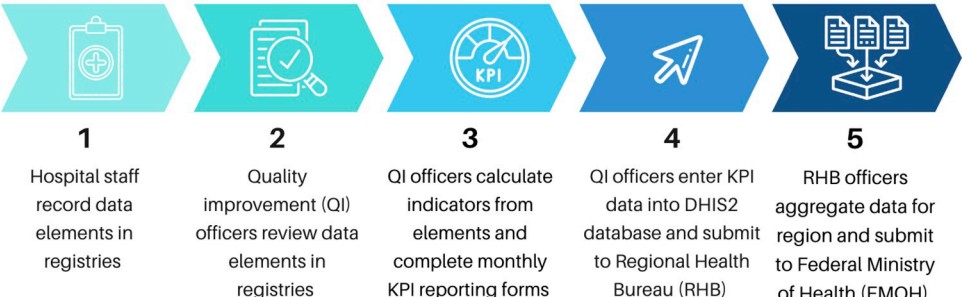

**Fig 1. Surgical indicator data flow.** Responsible data owner and reporting instrument at each step of the data collection and reporting process. KPI = Key Performance Indicators, DHIS2 = District Health Information Software 2.

purpose varied between hospitals. The KPI data is then entered into the electronic DHIS2 and uploaded to the Regional Health Bureau (RHB) for approval. KPI data at the RHB is aggregated from multiple hospitals together to describe the surgical practice in the region and submitted to the Federal Ministry of Health.

## Surgical key performance indicators

From registry data collected independently, values for the five KPIs of interest were calculated for each month from January to June 2022 (Tir to Sene 2014 in the Ethiopian Calendar). For all ten hospitals over six-months, average monthly surgical volume was 57 cases and SSC utilization was 93% (Table 3). Over one month, the mean surgical volumes were 54 and 59 cases per hospital in Amhara and Sidama, respectively. In Amhara, the majority of hospitals reflected the regional average. In Sidama, three primary hospitals performed less than 15 cases each on average per month, while the two general hospitals each performed between 55 and 214 operations on average per month. This extrapolates to a predicted rate of 50 to 712 operations per 100,000 population annually at primary hospitals in Amhara, 30 to 67 per 100,000 in Sidama primary hospitals, and 80 to 147 per 100,000 in Sidama general hospitals according to their determined catchment populations. When calculated using a standardized catchment population of 1 million for primary and general hospitals, there is a predictive rate of 35 to 121 operations per 100,000 population annually at primary hospitals in Amhara, 4.8 to 12 per 100,000 in Sidama primary hospitals, and 64 to 264 per 100,000 in Sidama general hospitals. Across all ten hospitals over the 6-month period there were thirteen post-operative mortality events, for a collective POMR of 0.38% (13/3399). For the same period, twenty-seven surgical site infections were reported for an overall inpatient SSI rate of 0.79% (27/3399). Five anesthetic adverse outcomes were documented for a collective AAO rate of 0.15% (5/3399).

## Data quality

Data quality was assessed based on two metrics: completeness and consistency. Our study found data completeness to be an average of 72% for the 10 hospitals, and 70% of hospitals had consistent data for the five surgical KPIs of interest. Verification factor calculations between registry and DHIS2 data, as seen in Fig 2, revealed inconsistencies (>10% difference) at nine total hospitals, commonly SSI (6/10) and surgical volume (5/10). Five hospitals demonstrated incomplete registry data, with the most commonly incomplete indicator data for AAO, SSC, and SSI. Across all five KPIs of interest, surgical volume was the most likely to be recorded within the registry and have inconsistencies between values recorded in the OR registry and

**Table 3. Comparison of registries to DHIS2 data for surgical KPIs.**

| | Median Surgical Volume | | | Median SSC | | | Total SSI | | | Total POMR | | | Total AAO | | |
|---|---|---|---|---|---|---|---|---|---|---|---|---|---|---|---|
| | Registry | DHIS2 | Inconsistent Months | KPI Forms[a] | DHIS2 | Inconsistent Months | Registry | DHIS2 | Inconsistent Months | Registry | DHIS2 | Inconsistent Months | Registry | DHIS2 | Inconsistent Months |
| **Sidama Hospitals** | | | | | | | | | | | | | | | |
| 1 | 4 (2–7) | 1.5 (0–7) | 50% | 56% (4.4–100%) | 56% (4.4–100%) | 0% | 0 | 0 | 0% | 0 | 0 | 0% | 0 | 0 | 0% |
| 2 | 55 (42–58) | 63.5 (60–67) | 100% | 100% | 100% | 0% | 0 | 0 | 0% | 2 | 1 | 17% | 0 | 0 | 0% |
| 3 | 213.5 (204–245) | 215.5 (205–234) | 0% | 100% (50–100%) | 100% (86–100%) | 33% | 3 | 3 | 33% | 11 | 0 | 83% | 0 | 0 | 0% |
| 4 | 10 (6–14) | 10 (6–16) | 33% | 100% | 100% | 0% | 0 | 1 | 17% | 0 | 0 | 0% | 0 | 0 | 0% |
| 5 | 8.5 (5–18) | 9 (6–18) | 50% | 100%[b] | 100% | 0% | 3 | 0 | 50% | 0 | 0 | 0% | 0 | 0 | 0% |
| **Amhara Hospitals** | | | | | | | | | | | | | | | |
| 6 | 97.5 (89–116) | 99.5 (89–116) | 100% | 100% | 100% | 0% | 14 | 8 | 50% | 0 | 0 | 0% | 0 | 0 | 0% |
| 7 | 26.5 (18–49) | 26.5 (18–49) | 100% | 100% | 100% | 0% | 0 | 0 | 0% | 0 | 0 | 0% | 5 | 0 | 33% |
| 8 | 52 (43–63) | 52 (42–63) | 100% | 100% | 100% | 0% | 3 | 0 | 33% | 0 | 0 | 0% | 0 | 0 | 0% |
| 9 | 46.5 (30–79) | 40.5 (28–53) | 50% | 87.5% (80–100%) | 100% | 67% | 3 | 1 | 50% | 1 | 0 | 17% | 0 | 0 | 0% |
| 10 | 41 (27–50) | 40.5 (28–51) | 100% | 100% | 100% | 0% | 1 | 0 | 17% | 0 | 0 | 0% | 0 | 0 | 0% |

Surgical Volume, SSC = Surgical Safety Checklist utilization, SSI = Surgical Site Infection, POMR = Perioperative Mortality Rate, AAO = Anesthesia Adverse Outcome. Median surgical volume with range for each hospital across the 6-month study period documented within hospital registries and DHIS2 report. Total surgery site infections, perioperative mortality events, and anesthesia adverse outcomes documented monthly in hospital registry and DHIS2 during the study period.

* [a]KPIs where monthly KPI forms were used as source data due to lack of registry data.

[b]Denotes that Hospital 5 SSC data was calculated from four months rather than six months as two months of KPI data were incomplete.

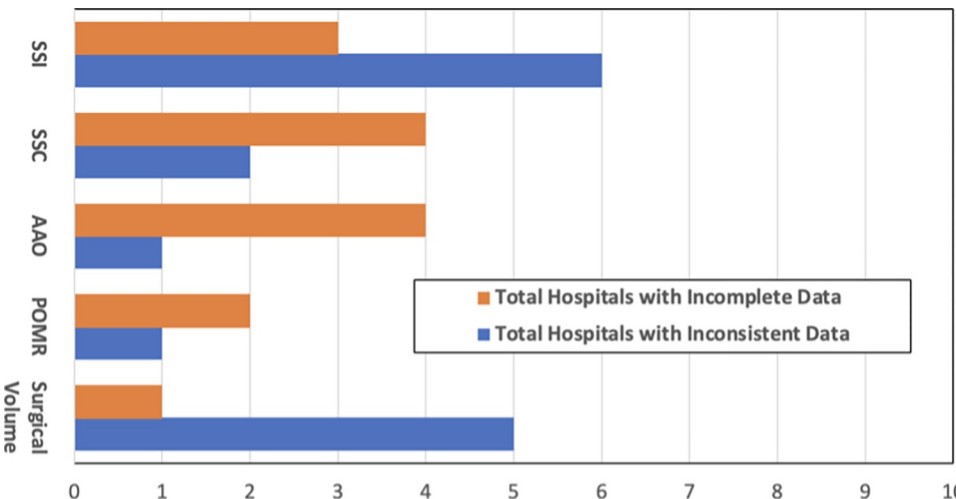

**Fig 2. Data quality assessment of five surgical indicators.** Number of hospitals out of 10 with inconsistent (blue) and incomplete (orange) data for each KPI of focus for the 6-month period. SSC = Surgical Safety Checklist utilization, SSI = Surgical Site Infection, POMR = Perioperative Mortality Rate, AAO = Anesthesia Adverse Outcome.

DHIS2 (Table 3). Of note, Sidama hospitals frequently had more inconsistent surgical volume data than Amhara hospitals.

After further investigation of inconsistencies between registry data and DHIS2 data, national data is revealing underreported rates (verification factor >1.1) for major surgical complications: SSI, AAO, and POMR. For POMR, 11 of the 13 events documented for all ten hospitals across 6-months were recorded at Hospital 3. However, this value does not correspond to the DHIS2 data which reported zero POMR events for Hospital 3 across the 6-months (Table 3). Furthermore, 1 POMR event observed at Hospital 9 for the reporting period did not transfer to the DHIS2 database but was present in the KPI reporting form. All of the hospitals who recorded any post-operative deaths (3 hospitals in total), had at least one month with inconsistent POMR data between registries and DHIS2. SSI rates were similarly underreported, with over half of the registry-documented surgical site infections at all ten hospitals over six months missing from the overall SSIs within DHIS2 (27/3399 vs 13/3399). SSI rates were inconsistent for seven out of ten hospitals for the evaluated period. For AAO, only Hospital 7 recorded 5 AAO events within their registry for the six-month period, but zero AAO events were present within DHIS2 for all ten hospitals over the study period. SSC data was mostly consistent between KPI forms and DHIS2 data, with the exception of Hospital 9 where median SSC differed by 12.5% and Hospital 3 where the range of SSC values differed by 36% but the median remained the same.

## Discussion

This study evaluated the process and quality of current surgical data practices in Ethiopia through five key performance indicators (KPIs) at ten hospitals in two regions over a 6-month period. The evaluation occurs five years after the creation and implementation of the surgical KPI data system in conjunction with Ethiopia's national strategic plan (SaLTS) to improve surgical and anesthesia care.

Principle findings include a largely uniform mechanism for data collection and reporting between facilities from hospital registries, to data collection forms, to the DHIS2 electronic database. However, the lack of standardization of data collection forms and staff practices

between each hospital may be a source of discrepancies in comparing outcomes from registries to DHIS2 data. While standardized registries have improved the collection of data in the setting of their origination, the development and implementation of standardized monthly KPI reporting forms across hospitals would improve consistency in recording and reporting. This will require a collaborative effort between policymakers, healthcare stakeholders, hospital employees, and non-governmental organizations.

Data quality was assessed based on two metrics: completeness and consistency which were found to be above 70% for the five KPIs at all 10 hospitals. Our study found data completeness was 72% on average for the 10 hospitals, and 70% of hospitals had consistent data for the five surgical KPIs of interest. Data quality varied based on the specific indicator. Surgical volume was largely complete throughout the data process, however at least 3 out of 10 hospitals had incomplete data for SSI, SSC, and AAO. Difficulties defining and tracking SSI and AAO perioperative complications are likely due to observed variability in recording forms and registries, reporting mechanisms across data flow, andwhich hospital provider is responsible for data collection.

Most inconsistencies are found for SSIs and surgical volume, with at least half of hospitals having inconsistent data for these two indicators. Some hospitals were not using standard registries, which could lead to missed data elements. For example, SSIs were highest in hospital 6 (n = 14), but only a little over half of these events made it into DHIS2 (n = 8). The initial data intervention included a SSI registry as part of the suite of tools for collecting and recording the indicators. Some hospitals had this SSI logbook to better capture this data, but there was variability in use of this system as opposed to using the hospital ward registries for recording infections.

The lack of or non-standardized data collection tools to transfer indicator components from the registries to DHIS2 also created a major gap in the link between these metrics. For example, Hospital 3 had the highest POMR (11 deaths) according to the registry data, but this was not accounted for in DHIS2 (0 deaths). We observed difficulty in accounting for surgical deaths from multiple different departments (ICU, multiple wards, and OR) which could explain this disparity, as deaths from the ICU were left out of the data process for POMR at this hospital. It also demonstrates the need for specifically trained personnel for data collection and entry, as well as routine data quality checks. Apart from the different areas of data management shortcomings due to lack of appropriate tools, there could be a high turnover of the already trained quality improvement officers. There should also be a uniform understanding of the primary data owners and the QI officers.

There were overall relatively low rates of complications such as SSI (0.79%), AAO (0.15%), and POMR (0.38%). It is possible that the data does not capture the full reality as outcomes are limited to inpatient events. Additionally, many of the hospitals, especially primary hospitals in Sidama, had lower surgical volume and cases were limited to cesarean-sections, which likely contributes to the lack of significant complications in these settings. An accurate assessment of the surgical system is dependent on these indicators as metrics of service delivery and quality of care. As the surgical system in Ethiopia continues to grow, it is imperative the quality of data is a true reflection of hospital operations. Given these rates of surgical complications may be lower than expected, a more accurate system for these indicators will allow for tracking improvements in access to and quality of surgical care.

## Surgical site infection

For SSI, our study estimated a value of 0.79% (27/3399) across 10 hospitals, lower than the 2% value recorded by the Network for Perioperative and Critical Care (N4PCC) 2021 registry [25]

and the 10.2% described in the African Surgical Outcomes Study [11]. Specifically for SSI, patients are often discharged before they will develop signs and symptoms of infection. There is currently no uniform mechanism to enumerate outpatient SSIs.

### Anesthesia adverse outcome

Similarly for AAO, our dataset of 3399 surgeries yielded only 5 AAO events for a rate of 0.15%. This is considerably lower than the N4PCC 2021 registry value of 3.1% [25] and closer to the AAO reported value of 0.94% in Zimbabwe [26]. The use of a multicenter cloud-based registry, in-hospital coordinators and data trainers, and utilization of an offline mobile application for data collection most likely improved reporting compliance and identification of adverse events, leading to increased AAO capture rates in the N4PCC study [25]. Variation in AAO rates reported can also be attributed to variations in KPI definitions, as in Zimbabwe where an AAO is defined as a critical incident and includes nausea or post-operative pain [26]. Furthermore for AAO, it may be difficult to detect and diagnose complications such as high spinal anesthesia. This was also the indicator with the greatest variability in recording mechanisms and data ownership, likely accounting for the very low AAO, with only one hospital reporting these events. Additional training for knowledge and skills may be needed in this case.

### Peri-operative mortality rate

Our study found a lower POMR at 0.38% (13/3399) than the 0.9% (9/1000) found in the N4PCC 2021 registry surveyed at 4 Ethiopian hospitals [25]. This disparity may be due to the indicators being limited to only inpatient events, thus POMR may be lower than expected as deaths for KPIs are registered only before discharge. There were also some deaths recorded in registries, which were not reported in the national data as mentioned previously. Further external mortality audits for these complications including chart reviews and periodical follow-up surveys, as well as a robust death registry, are recommended to understand the true state of surgical outcomes in Ethiopian hospitals.

### Surgical volume

Calculated annual surgical volume for Sidama general hospitals is 80 to 147 operations per 100,000, 50 to 712 operations per 100,000 population at primary hospitals in Amhara, and 30 to 67 per 100,000 in Sidama primary hospitals. In our dataset, surgical volume at all ten hospitals has not yet reached the LCoGS 2030 target of 5000 surgeries per 100,000 population annually [2]. Study findings of surgical volume for Sidama general hospitals (Jan- June 2022) are consistent with Meshesha *et al.*'s surgical volume of 289 per 100,000 over a 90-day interval (Sept 2020—May 2021) for general hospitals [27]. Our study calculates higher annual surgical volumes for Amhara and Sidama primary hospitals than the 37 per 100,000 found for primary health care units over a 90-day interval. The Amhara primary hospitals showed an overall increase in their surgical volume compared to the previous study during the data intervention in 2018 [28], which may be due to both growth in surgical capacity and population needs.

### Surgical safety checklist

Estimated SSC values were 93% for 3399 surgeries, consistent with the 92.1% found in the N4PCC registry for 1595 surgical cases [25] and higher than the 67.6% rate reported by Sibhatu *et. al.* 2022 [29]. Our study determined 60% of audited hospitals had complete SSC data, consistent with the 60.8% of 659 checklists the latter study found to be filled completely and correctly. The barriers to data collection identified at ten hospitals within the Amhara and Sidama

region are consistent with the findings of previous investigations into data quality within the Ethiopian healthcare system. The overreporting of certain surgical KPIs, namely SSC and underreporting metrics of illness or mortality was concurrent with findings from Mekebo *et. al.* and other studies [28, 30, 31]. Our observation of data flow processes demonstrated that SSC values are calculated by QI officers by selecting a sample of ten patient charts for review. At certain institutions, stop-gap mechanisms were implemented to prevent the progression of patient charts to the Inpatient Admission office (IPAD) until all forms including the SSC were completed. Such processes might explain the disparity Sibhatu *et. al* found between the DHIS2 SSC value of 81% and the externally audited 60.8% retrospectively calculated from patient charts [27]. High rates of surgical data inconsistency were a major result of this research and are likely due to lack of consistent KPI reporting forms, and the considerable burden of aggregating substantive data elements into complex indicators as discussed previously by Adane *et al* 2021 [32]. Our observations were in alignment with other challenges with data collection in an Ethiopian setting previously discussed including turnover of hospital staff, poor understanding of data processes, modification, or manipulation of data to compensate for the lack of data, and technological issues with DHIS2 [28, 30].

## Limitations

National implications of this study are limited by the small sample size (ten hospitals) and limited study period (six months). Hospital selection was also limited to facilities with prior involvement in Safe Surgery 2020. This limitation could potentially impact the generalizability of these findings to the current surgical landscape in Ethiopia, as system changes are ongoing and interregional differences are likely. The investigators included two regions within Ethiopia as well as both primary and general hospitals as countermeasures to the limited sample size and period. Selecting hospitals involved in the prior surgical data strengthening program (in Amhara) and those not (in Sidama) was also employed to expose potential differences in data practices. Follow-up investigation using qualitative methods, specifically surveying data collectors and data managers at participating hospitals and regions, will provide further context around the findings. Future studies are also planned to expand this initial investigation of key performance indicator data in Ethiopia to a larger number of hospitals and regions.

Additionally, the use of registry data as primary source data generates the risk that some surgical indicators not recorded in the registry were left out of our analysis. Using registry data rather than performing a retrospective review of surgical patient charts could underestimate issues with data quality. Mortality audits, or complication audits, are another potential retrospective mechanism to capture more complete data for these outcomes, requiring access to patient charts [33, 34] Future studies incorporating alternative mechanisms, such as those described, will contribute important information on the accuracy and reliability of registry data recorded during the patient's hospital course.

Another source of data quality inconsistency could be attributed to reporting period variability between hospitals. This study utilized DHIS2 reporting periods for data auditing. The implications of variable reporting periods are issues with accuracy and comparison interregionally and between hospitals. An under or over-estimation of the KPI data from month to month could occur as well as the omission or double counting of a recordable event during registry review by the KPI data officers. Our study recommends stricter enforcement of DHIS2 reporting periods within hospital data collection systems including but not limited to emphasizing correct reporting periods during KPI training of hospital personnel and providing standardized registries and KPI reporting forms organized by the correct reporting period with clear cut off dates to enhance the data system.

Other limitations include the inability to quantitatively assess the level of data correction and manipulation by hospital staff following recording in the registry and prior to DHIS2 input. Concern over the consequences of reporting values too high or low for a given indicator likely compound this issue, generating issues with data reliability in turn Careful consideration of the procedures used to review registry data, the consequences associated with reporting poor metrics, and the use of this data to allocate government resources to hospitals would likely improve overall data quality.

The study team noted blank entries were often used to denote "0" findings within data forms, challenging the ability to assess data completeness versus a true lack of recordable events for the month. From the perspective of the KPI focal person, whose role it is to review and assimilate the primary source registry data into KPI monthly reporting forms, this limitation affects follow-up with data collectors regarding potential missed events. Our study team recommends universal guidelines for all hospital staff recording and reviewing KPI data, including indicating no recordable events with "0" rather than a blank entry to create a clearer report. Encouraging data ownership and responsibility by primary recorders i.e. Head OR nurse, Liaison officer, etc. would encourage frequent review of recorded data for clarity and serve as a countermeasure for this limitation.

Finally, this study was unable to include hospitals involved in previous data quality interventions in Tigray because of ongoing conflict and travel restrictions. This limits the comparability of this study to prior works by the study team. However, the inclusion of Amhara region hospitals within the current work and prior works improves the comprehensiveness of the findings. Ultimately, addressing data gaps in conflict-affected regions remains a significant challenge. While the DHIS2 database would allow for remote viewing of KPI aggregated data, the collection of this data is highly dependent on local stability, including the existence of permanent infrastructure and electronic access.

## Recommendations

Our study recommends a regular, multi-level supervision mechanism focused on identifying and addressing data inconsistencies to ensure standardized data processes across all hospitals. Routine indicator quality inspections at each level of the system are essential to address the currently described issues.

At the national level, an emphasis on the importance of data quality while simultaneously ensuring no penalty for suboptimal outcomes is imperative. Incentives for consistent data quality (hospital recognition at the national level) as opposed to low complication rates would aid in the cultural transformation of this ideal. Additional proposed solutions include the integration of an electronic health system, which is currently in its early stages in the country at select hospitals, to have real-time data entry and minimize the opportunity for error. Reinvigorating the monitoring and evaluation arm of the SaLTS team is another potential avenue for improvement. While there are systems for evaluating national health system data quality, ultimately the emphasis on accuracy needs to extend to the regional and hospital level.

The study team similarly recommends at the regional level, the implementation of permanent survey teams to decrease the risk of errors and improve data quality and utilization with regular supervision and external audits. This system is currently in place in some regions, including Sidama, and initial feedback from this study has been used for this purpose. Having identified areas of data inconsistency, the Sidama regional office has worked with local hospital teams to improve the way in which these complications (specifically POMR) are recorded and reported. While the current system emphasizes data completeness on a monthly basis, quarterly audits of consistency are an additional requirement to enhance indicator capture. While

these audits can be resource-intensive, the upfront investment is likely to lead to more sustainable, accurate health data which can ultimately improve the health system. Frequent training and incentives for quality officer retention would improve the consistency of this oversight.

At the hospital level, more consistent quality improvement programs to emphasize the relevance of these indicators can provide an opportunity to focus on surgical complications as areas for better teamwork, while increasing data capture as a byproduct. At one hospital, a department-wide emphasis on reducing surgical site infections, with large publicly visible posters denoting monthly progress helped to improve more accurate collection of this variable. Since the surgical team was motivated to improve this indicator, they were also keen to precisely measure monthly progress. Ultimately, hospital data quality officers will be the ones performing the most in-depth quality assessments of each department's data on at least a monthly basis as required regionally and nationally. An additional opportunity is for more regular training sessions involving all participants in the data flow process at the hospital, not limited to data quality officers.

## Conclusion

This study demonstrates the ongoing challenges to accurate surgical data collection in a resource-limited setting. Scarce human resources, limited time for data collection, reporting, and auditing, interrupted internet connectivity, and a high volume of patient care needs are all threats to surgical data quality. Five years after the implementation of Ethiopia's surgical indicators as part of their national surgical and anesthesia strategic plan, there remains difficulty with capturing the true value of surgical complications such as infections and perioperative mortality. This national system for data collection and reporting was a momentous step forward in understanding the current state of surgery in Ethiopia and tracking progress towards measurable goals in improving surgical capacity and quality. However, more unified mechanisms for data transfer, ongoing training for quality officers and health professionals, as well as regular audits for data consistency are needed to improve the current structure. Capturing outpatient surgical outcomes as well as the development of an electronic registry, which is currently in development, will lead to improvements in monitoring and improving surgical capacity and quality in Ethiopia while addressing several of the aforementioned challenges in this setting [21, 25].

## Supporting information

**S1 Checklist. Inclusivity in global research.**
(DOCX)

**S1 Data. Data repository.**
(XLSX)

## Acknowledgments

We thank the Ethiopian Federal Ministry of Health, Sidama Regional Health Bureau, Amhara Regional Health Bureau, and MCW Office of Global Health for their support and collaboration.

## Author Contributions

**Conceptualization:** Kayleigh R. Cook, Zebenay B. Zeleke, Ephrem Gebrehana, Daniel Burssa, Atkilt Michael, Andualem Beyene, Katherine R. Iverson.

**Data curation:** Kayleigh R. Cook, Zebenay B. Zeleke, Ephrem Gebrehana, Bantalem Yeshanew, Atkilt Michael, Yoseph Tediso, Andualem Beyene, Katherine R. Iverson.

**Formal analysis:** Kayleigh R. Cook, Andualem Beyene, Katherine R. Iverson.

**Funding acquisition:** Katherine R. Iverson.

**Investigation:** Kayleigh R. Cook, Zebenay B. Zeleke, Ephrem Gebrehana, Andualem Beyene, Katherine R. Iverson.

**Methodology:** Kayleigh R. Cook, Zebenay B. Zeleke, Ephrem Gebrehana, Yoseph Tediso, Andualem Beyene, Katherine R. Iverson.

**Project administration:** Zebenay B. Zeleke, Andualem Beyene, Katherine R. Iverson.

**Resources:** Zebenay B. Zeleke, Daniel Burssa, Atkilt Michael, Yoseph Tediso, Chris Dodgion, Andualem Beyene, Katherine R. Iverson.

**Software:** Atkilt Michael, Katherine R. Iverson.

**Supervision:** Zebenay B. Zeleke, Ephrem Gebrehana, Chris Dodgion, Andualem Beyene, Katherine R. Iverson.

**Validation:** Andualem Beyene, Katherine R. Iverson.

**Visualization:** Katherine R. Iverson.

**Writing – original draft:** Kayleigh R. Cook, Andualem Beyene, Katherine R. Iverson.

**Writing – review & editing:** Kayleigh R. Cook, Zebenay B. Zeleke, Ephrem Gebrehana, Daniel Burssa, Bantalem Yeshanew, Atkilt Michael, Yoseph Tediso, Taylor Jaraczewski, Chris Dodgion, Andualem Beyene, Katherine R. Iverson.

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
