## [Decision Letter · Decision Letter 0]

8 Jan 2024

PGPH-D-23-02067

Quality and sustainability of Ethiopia’s national surgical indicators

Dear Dr. Cook,

Thank you for submitting your manuscript to PLOS Global Public Health. After careful consideration, we feel that it has merit but does not fully meet PLOS Global Public Health’s publication criteria as it currently stands. Therefore, we invite you to submit a revised version of the manuscript that addresses the points raised during the review process.

Thank you for this well written manuscript that demonstrates the ongoing challenges to accurate surgical data collection in a resource-limited setting. This material should be published and will be insightful both for Ethiopia and other similar contexts in sub-Saharan Africa. The study team must be commended for the focus on equity in authorship that comes through in the mix of first and senior authors, so that this appears to be truly collaborative, and a win-win partnership between LMIC and HIC partners.

Kindly add in more detail on the ethics approval/Institutional Review Board documentation into the text. Adding the numbers on the ethics approval/certificate makes for a more transparent report. Kindly provide these details for the Medical College of Wisconsin and the Ethiopian Public Health Institute.

**Also add in a clearer data availability statement, making limitations to public data sharing clear in the Data Availability Statement at the time of submission. **"PLOS journals require authors to make all data necessary to replicate their study’s findings **publicly** available without restriction **at the time of publication.** When specific legal or ethical restrictions prohibit public sharing of a data set, authors must indicate how others may obtain access to the data. This data refers to 'the minimal dataset'  required to replicate all study findings reported in the article, as well as related metadata and methods." The latter seems to be the case with this work,but the data availability statement will be seen by all, including the fact that it should not be published.

PLOS recognizes that, ***i**n** some instances, authors may not be able to make their underlying data set publicly available for legal or ethical reasons***. This data policy does not overrule local regulations, legislation or ethical frameworks. ***Where these frameworks prevent or limit data release, authors must make these limitations clear in the Data Availability Statement at the time of submission***. Acceptable restrictions on public data sharing are detailed on the PLOS site. Also note it is not acceptable for an author to be the sole named individual responsible for ensuring data access.

Please address all the reviewer's concerns concerning the limitations and recommendations. This is required for acceptance. Both reviewers of this work are very knowledgeable about the local context and have suggested thoughts around the recommendations and improving the limitations section. None of the concerns were major, but applying these thoughts can help improve the draft, and clarify concerns that readers might raise. Overall, this is very clear and well done.

We look forward to receiving your revised manuscript.

Kind regards,

Barnabas Tobi Alayande

Academic Editor

Journal Requirements:

2. Please amend your online Financial Disclosure statement. If you did not receive any funding for this study, please simply state: “The authors received no specific funding for this work.”

3. Please update your online Competing Interests statement. If you have no competing interests to declare, please state: “The authors have declared that no competing interests exist.”

4. Some material included in your submission may be copyrighted. According to PLOS’s copyright policy, authors who use figures or other material (e.g., graphics, clipart, maps) from another author or copyright holder must demonstrate or obtain permission to publish this material under the Creative Commons Attribution 4.0 International (CC BY 4.0) License used by PLOS journals. Please closely review the details of PLOS’s copyright requirements here: PLOS Licenses and Copyright. If you need to request permissions from a copyright holder, you may use PLOS's Copyright Content Permission form.

Potential Copyright Issues:

Figure 1: Please confirm whether you drew the images / clip-art within the figure panels by hand. If you did not draw the images, please provide (a) a link to the source of the images or icons and their license / terms of use; or (b) written permission from the copyright holder to publish the images or icons under our CC-BY 4.0 license. Alternatively, you may replace the images with open source alternatives. See these open source resources you may use to replace images / clip-art:

- https://openclipart.org/

Additional Editor Comments (if provided):

Reviewers' comments:

Reviewer's Responses to Questions

**Comments to the Author**

1. Does this manuscript meet PLOS Global Public Health’s publication criteria? Is the manuscript technically sound, and do the data support the conclusions? The manuscript must describe methodologically and ethically rigorous research with conclusions that are appropriately drawn based on the data presented.

Reviewer #1: Yes

Reviewer #2: Yes

2. Has the statistical analysis been performed appropriately and rigorously?

Reviewer #1: Yes

Reviewer #2: Yes

3. Have the authors made all data underlying the findings in their manuscript fully available (please refer to the Data Availability Statement at the start of the manuscript PDF file)?

Reviewer #1: Yes

Reviewer #2: Yes

4. Is the manuscript presented in an intelligible fashion and written in standard English?

Reviewer #1: Yes

Reviewer #2: Yes

5. Review Comments to the Author

Reviewer #1: Based on the Saving Lives through Safe Surgery (SaLTS) initiative launched by the Ethiopian Federal Ministry of Health, introducing 15 key performance indicators (KPIs) to standardize data practices, the study aims to evaluate KPI data collection practices and assess data quality across 10 hospitals in two regions. The investigation compares five selected key indicators, including Surgical Volume, Perioperative Mortality Rate (POMR), Adverse Anesthetic Outcome (AAO), Surgical Site Infection (SSI), and Safe Surgery Checklist (SSC) Utilization. Findings reveal discrepancies in data consistency, with incomplete information in registries and challenges related to non-standardized reporting forms. Facilitators include the continued use of registries, while barriers involve variable indicator recording and delayed indicator rollout. Recommendations focus on continuous quality checks and capacity building to enhance data management for improved healthcare systems.

Overall, the study is relevant, and well written with specific comments and minor adjustment suggestions described below for the authos' consideration.

Introduction: It provides a good overview of the global significance of surgical care data and connects it well into available data within the country.

Methods: The study design, data analysis, and method of measure are articulated clearly. The rationale with selection of study sites is well explained, however, information on the reason behind the chosen time frame (Jan-June 2022) should be further clarified and justified.

Results: The main findings are presented clearly with effective summarization through appropriate table and figures.

Discussion: It provides insightful points, drawing comparison with findings from similar studies done in other countries.

Limitations: The study offers a comprehensive overview of its limitations. However, it would be beneficial to further elaborate on any countermeasures implemented by the study team to mitigate potential biases encountered, if any.

Recommendations: While the need for qualitative assessment was acknowledged as a limitation in the study, incorporating a recommendation for future studies to explore the reasons behind data discrepancies would significantly contribute to devising targeted interventions for improving the quality of surgical care data.

Conclusion: Major points and messages are effectively summarized.

In summary, the study provides valuable insights into KPI data collection practices within the SaLTS initiative underscoring the need for continuous quality improvement efforts to ensure a robust surgical database in the country and provides a strong foundation for future research.

Reviewer #2: I have carefully reviewed this scientific paper on the quality and sustainability of Ethiopia’s national surgical care indicators.

1. Relevance and Significance- The paper effectively addresses a critical area of concern in surgical care in LMICs particularly assessing the nuance of surgical indicators in terms of sustainability integrity, gaps, and areas of potential improvement. The relevance of the topic is evident given the global movement and effort to improve access to quality, safe, timely, and appropriate surgical care and the need for data driven decision making.

2. Methodology- The methodology section was well detailed, providing data collection, analysis, definitions of framework, completeness etc. Key concerns re: data integrity were also raised although I recommend a more robust discussion of the limitations (which authors have done later in the limitation section).

3. Data Presentation and Analysis- the data presentation's is generally clear and analysis is appropriately conducted. Some visual aids such as tables and charts were used which is good. However, a more detailed graph or analysis in writing which tries to break down the key points within the table would make the information more absorbable. Even further, I would have also loved to see a more robust explanation of the relationship between the metrics, KPI and/or indices and what that entails, both for the paper and for the overall quality of care and in terms of addressing the desired outcome as well as how that impacts the analysis and the overall service delivery and patient care (even if it gets elaborated in the discussion segment). They can indeed try a more sophisticated analysis on Excel which is what they have tried but I generally recommend trying Tableu as a great data visualization tool as it gives the non expert reader a more digestible, high quality, and summarized outcome. This will also hopefully aid high level stakeholders such as decision makers at FMOH an efficient way to aid future policy making and advocacy efforts.

4. Discussions and implications-

A) Standardization of data collection practices: The authors rightly highlight the uniformity in data collection mechanisms but note discrepancies due to non-standardized data collection forms and practices. To address this, it is crucial to recommend the development and implementation of standardized data collection tools across all hospitals. Emphasize the importance of a collaborative effort involving healthcare stakeholders, policymakers, and professional organizations to establish and endorse standardized forms, ensuring consistency in recording and reporting across facilities.

B. Data Quality and Completeness: While the study finds data completeness and consistency above 70% for the five KPIs, making recommending specific measures to enhance data quality would . This making granular suggestions and recommendations may enhance their work. This could be regular training sessions for healthcare staff involved in data collection, routine data quality checks, and the utilization of technology for real-time data entry to minimize errors. Maybe even propose the establishment of a centralized oversight body responsible for monitoring and ensuring the quality and completeness of surgical data at the national level. Similarly, for the below points

C) Addressing inconsistencies in surgical volume and complications

D) Training and retention of quality improvement officers

E) External audits and continuous quality inspections

F) Recommendation for policy makers

In conclusion, the authors' identification of gaps in surgical data practices is commendable, and the discussion section can be strengthened by providing nuanced and actionable recommendations for addressing these challenges. The proposed enhancements aim to guide policymakers and healthcare stakeholders in implementing practical measures to improve the overall quality and reliability of surgical data in Ethiopia.

5. Limitations

A. Sample Size and Study Period: The acknowledgement of the small sample size (10 hospitals) and the limited study period (six months) is appropriate. However, suggest expanding on the potential impact of these limitations on the generalizability of the findings. Discuss whether the findings can be considered representative of the broader healthcare landscape in Ethiopia.

B. Data Source and Quality: The mention of using registry data as the primary source is well-noted. Suggest discussing the potential biases introduced by relying solely on registry data, such as the underestimation of data quality issues. Encourage a deeper exploration of alternative data sources, like mortality audits, to supplement and validate the findings.

C. Reporting Period Variability: The acknowledgment of reporting period variability is crucial. Elaborate on how this variability might affect the accuracy and comparability of the data. Provide recommendations on standardizing reporting periods across hospitals to enhance the consistency of data collection and reporting.

D.Data Correction and Manipulation: Address the limitations associated with the inability to quantitatively assess the extent of data correction and manipulation by hospital staff. Discuss the potential implications of data correction on the reliability of the reported indicators and suggest strategies for minimizing such corrections, ensuring data accuracy.

E. Blank Entries and Data Completeness: Discuss the challenges posed by the use of blank entries to denote "0" findings within data forms. Provide recommendations for improving data completeness and suggest alternative methods for capturing zero values without compromising the ability to assess data completeness accurately.

F. Exclusion of Hospitals in Tigray: The exclusion of hospitals in Tigray due to ongoing conflict and travel restrictions is a significant limitation. Discuss the potential impact of this exclusion on the comprehensiveness of the study and suggest strategies for addressing data gaps in conflict-affected regions.

6. Recommendations

Supervision Mechanism and Data Consistency:

Expand on the recommendation for a regular supervision mechanism to address data inconsistencies. Provide details on the structure and frequency of such supervision, emphasizing its role in enhancing data quality. Consider discussing the potential challenges in implementing and sustaining this mechanism.

National Emphasis on Data Quality:

Strengthen the recommendation for a national emphasis on data quality. Discuss specific strategies for promoting a culture of data quality at the national level, considering the unique challenges faced in a resource-limited setting. Highlight the importance of balancing the pursuit of high-quality data with avoiding punitive measures for suboptimal outcomes.

Regional Survey Teams:

Further elaborate on the recommendation for implementing permanent survey teams at the regional level. Discuss the anticipated impact of such teams on reducing errors and improving data quality. Provide examples of successful implementation in regions where this system is already in place.

Hospital-Level Quality Improvement Programs:

Enhance the recommendation for more consistent quality improvement programs at the hospital level. Discuss the potential benefits of these programs in fostering better teamwork and increasing data capture. Provide practical suggestions for incorporating these programs into routine hospital practices.

Conclusion:

Challenges in Surgical Data Collection:

The conclusion effectively highlights the persistent challenges in accurate surgical data collection in a resource-limited setting. Consider expanding on specific challenges faced by healthcare professionals and institutions, providing context for readers unfamiliar with the intricacies of data collection in such settings.

Impact of National System:

Emphasize the impact of the national system for data collection and reporting, highlighting its importance to ongoing progress. Discuss the successes and shortcomings of the existing system, acknowledging its role in laying the foundation for future improvements.

Unified Mechanisms and Continuing Improvement:

Reinforce the need for more unified mechanisms for data transfer, ongoing training, and regular audits to improve the current data structure. Elaborate on how these recommendations align with the broader goals of Ethiopia's national surgical and anesthesia strategic plan.

Future Developments:

Strengthen the conclusion by discussing future developments, such as the ongoing development of an electronic registry. Explore how these advancements could contribute to addressing current challenges and fostering continuous improvement in tracking surgical capacity and quality in Ethiopia.

In summary, while the limitations and recommendations provided are insightful, further elaboration on the potential implications of these limitations and more specific details in the recommendations would strengthen the overall depth and nuance of the discussion. Additionally, enhancing the conclusion with a forward-looking perspective on future developments could provide a more comprehensive and impactful closure to the study.

6. PLOS authors have the option to publish the peer review history of their article (what does this mean?). If published, this will include your full peer review and any attached files.

**Do you want your identity to be public for this peer review?** For information about this choice, including consent withdrawal, please see our Privacy Policy.

Reviewer #1: No

Reviewer #2: **Yes: **Selam Degu

---

## [Editor Report · Decision Letter 1]

14 Feb 2024

Quality and sustainability of Ethiopia’s national surgical indicators

PGPH-D-23-02067R1

Dear Ms Cook,

We are pleased to inform you that your manuscript 'Quality and sustainability of Ethiopia’s national surgical indicators' has been provisionally accepted for publication in PLOS Global Public Health.

Best regards,

Barnabas Tobi Alayande

Academic Editor